# Wavelet-Based Topological Loss for Low-Light Image Denoising

**DOI:** 10.3390/s25072047

**Published:** 2025-03-25

**Authors:** Alexandra Malyugina, Nantheera Anantrasirichai, David Bull

**Affiliations:** Visual Information Laboratory, University of Bristol, Bristol BS1 5DD, UK; n.anantrasirichai@bristol.ac.uk (N.A.); dave.bull@bristol.ac.uk (D.B.)

**Keywords:** image denoising, persistent homology, topological data analysis, loss function, wavelet transform

## Abstract

Despite significant advances in image denoising, most algorithms rely on supervised learning, with their performance largely dependent on the quality and diversity of training data. It is widely assumed that digital image distortions are caused by spatially invariant Additive White Gaussian Noise (AWGN). However, the analysis of real-world data suggests that this assumption is invalid. Therefore, this paper tackles image corruption by real noise, providing a framework to capture and utilise the underlying structural information of an image along with the spatial information conventionally used for deep learning tasks. We propose a novel denoising loss function that incorporates topological invariants and is informed by textural information extracted from the image wavelet domain. The effectiveness of this proposed method was evaluated by training state-of-the-art denoising models on the BVI-Lowlight dataset, which features a wide range of real noise distortions. Adding a topological term to common loss functions leads to a significant increase in the LPIPS (Learned Perceptual Image Patch Similarity) metric, with the improvement reaching up to 25%. The results indicate that the proposed loss function enables neural networks to learn noise characteristics better. We demonstrate that they can consequently extract the topological features of noise-free images, resulting in enhanced contrast and preserved textural information.

## 1. Introduction

Denoising is a common task in processing low-quality image data and is an important component in the processing pipeline for cases where acquisition is performed under low-light conditions. Applications include object detection and object tracking in autonomous vehicles, biomedical image analysis, natural history filmmaking and consumer video production. The primary objective of image denoising is to eliminate the noise present while preserving the original features. However, in practice, denoising poses an inverse problem with intrinsically ambiguous solutions, and hence always results in a trade-off between signal distortion and noise removal. Moreover, denoising methods should not introduce new artefacts, for example over smoothing (blurring), that can degrade the visual quality of the image.

Conventional denoising algorithms (pre-machine learning), such as Non-Local Means (NLMs) [1] and Block-Matching and 3D filtering (BM3D) [2], rely on mathematical models and involve pre-defined filters to remove noise from images. Latterly, state-of-the-art denoising methods have exploited deep learning-based approaches [3]. Some methods have adopted a self-supervised learning framework to address the issue of unavailable ground truth data, e.g., [4,5,6,7,8]. However, the performance of these methods is limited, particularly for low-light content where, for example, edges are significantly distorted due to the presence of noise [9]. The best-performing methods hence rely on supervised learning, where models are trained either by applying synthetic noise to the clean images or by creating pseudo ground truth data from real noise images [10,11,12,13]. A comprehensive review of learning-based image denoising is available in [14].

Deep learning approaches are typically developed by designing a network architecture and defining a loss function. In this paper, we focus on the latter. The loss function plays a critical role in measuring the performance of the neural network, and the goal of training is to minimise this loss. Through the backpropagation algorithm, the loss function provides feedback to the neural network by computing its gradient, which is then used to update the network’s weights. Commonly used loss functions for image denoising include $\ell_{1}$, $\ell_{2}$, Charbonnier, SSIM, Laplacian loss, and a combination of these [12]. Other losses have also been employed, including perceptual loss, wavelet-based loss, gradient loss and total variation loss, but no performance improvement has been reported with these methods [9]. Despite significant efforts to improve the performance of denoising methods, their results still typically exhibit distortion artefacts or lost signal information. Moreover, there are no reports that analyse how the loss function influences noise removal for different content classes.

In this paper, we introduce a novel loss function, combining local low-level features and topological features extracted from the image. The denoising model is trained to be selective with respect to the textured areas of the image, resulting in improvements in contrast and performance on edges. Our proposed loss function is based on our previous work in [15]; we further enhance performance by using wavelets to provide different content classes, such as homogeneous regions, edges, structures, and textures with different weights. We then create a weight map to apply the weighed loss to each category. This is applied at the pixel level resulting in enhanced sharpness and spatial contrast in the output image. Additionally, we calculate topological invariants of the whole image instead of patches (as reported in [15]). We compare the performance of the new loss function across popular denoising architectures, revealing that adding a topological term to conventional loss functions helps to improve subjective results by enhancing contrast while preserving texture. In conclusion, the main contributions of our paper are
A novel denoising loss function that incorporates topological invariants and leverages textural information from the image wavelet domain. The proposed framework effectively integrates both the underlying structural information and spatial characteristics of images.Evaluation of the proposed loss function with multiple denoising models on the BVI-Lowlight dataset, which is specifically designed for real noise distortions in low-light conditions.

The remainder of this paper is organised as follows. The problem statement and existing solutions are presented in Section 2. Details of topological invariants and the proposed loss function for image denoising are described in Section 3 and Section 4, respectively. The performance of the method is evaluated in Section 5, followed by the conclusions in Section 6.

## 2. Image Denoising Problem and Current Solutions

### 2.1. Image Denoising Problem

Given an image $I^{N}$ corrupted by noise *N*, the goal is to obtain a denoised image $I^{O}$ that accurately represents the underlying clean image $I^{C}$. Mathematically, this can be expressed as follows:$$I^{O}=f_{\theta }\left(I^{N}\right),$$
where $f_{\theta }$ represents the denoising function or model that takes the noisy image as input and produces the denoised image as output. The noisy image $I^{N}$ in this case is the result of some function (e.g., signal-dependent noise) of $I^{C}$:$$I^{N}=Noise\left(I^{C}\right)$$

The task of denoising can then be formulated as an optimisation problem, where one searches for an optimal parameter set $\theta ={\left\{\theta_{i}\right\}}_{i\in I}$ for a denoising model $f_{\theta }$ through minimising loss function L:$$\hat{\theta }=\arg\min_{\theta }L\left(f_{\theta }\left(I^{N}\right),I^{C}\right).$$

The denoising process aims to estimate the clean image $I^{C}$ by suppressing or removing the noise component from the noisy image $I^{N}$. This involves leveraging statistical properties of the noise and exploiting the underlying structure and features present in the image.

### 2.2. Current Image Denoising Methods

Denoising methods can be categorised based on the domain they operate on and the type of approach employed.

In terms of the domain, denoising methods can be classified into spatial domain, transform domain, and hybrid (spatio-domain) methods. Spatial domain methods operate directly on the pixel values of an image. They include local methods such as mean filters, order-statistic filters, and adaptive filters. Non-local methods, like non-local means (NLMs), take advantage of image self-similarity for denoising. Techniques such as steering kernel regression (SKR) [16] and Wiener filtering [17] have been proposed to improve the performance of spatial filters in structured areas containing edges [18].

Transform domain methods involve representing image patches using orthogonal basic functions and adjusting coefficients to denoise the image. Wavelet-based denoising methods have gained popularity due to their sparsity and multiresolution properties. Thresholding techniques, such as VISUShrink [19] and BayesShrink [20], are commonly used in wavelet domain denoising [21,22,23,24,25].

Dual-domain methods combine spatial and transform domain filtering. The BM3D algorithm, proposed by Dabov et al. [26], utilises collaborative filtering in both the spatial and transform domains. It groups similar image patches, applies transform-domain shrinkage, and performs inverse transforms to obtain denoised images. Shape-adaptive versions of BM3D further improve the preservation of image details [2,27].

In terms of the type of approach, denoising methods can be divided into conventional and learning-based methods. Conventional methods typically employ handcrafted algorithms and heuristics to remove noise from images. These approaches include filters such as mean filters, median filters, order-statistic filters, and many of the above-mentioned techniques (NLM, BM3D, and thresholding).

Learning-based methods, such as sparse representation and dictionary learning, have been applied to image denoising. Techniques such as K-SVD [28] and locally learned dictionaries (K-LLD) [29] utilise sparse coding and clustering techniques to represent image patches. The LSSC method proposed by Mairal et al. [30] combines local and non-local sparsity within a neighbourhood. Deep learning techniques, specifically convolutional neural networks (CNNs), have shown remarkable performance in image denoising. DnCNN, introduced by Zhang et al. [10], is a deep neural network architecture based on residual learning that has achieved excellent results. One of the state-of-the-art image denoising methods, RIDNET [11], utilises attention mechanisms with short and long connections to learn both local and global features. More recent learning-based denoisers combine transformers with UNet architectures, which enhances feature extraction and hierarchical representation capabilities [31]. The performance is comparable to RDUNet [32], where residual dense network is employed through UNet architectures.

It is worth noting that there is a distinction between supervised and unsupervised deep learning denoising. Supervised methods rely on ground truth images for training, while unsupervised methods aim to denoise images without ground truth information. Deep Image Prior [6], Noise2Noise [33], and Blind2Unblind [7] are examples of unsupervised deep learning approaches, where networks are trained using only noisy input images. Conventional techniques often suffer from insufficient adaptability due to their reliance on fixed filters or predefined parameters, while learning-based denoising methods face challenges related to data requirements, generalisation, and handling of new noise types. Both conventional and learning-based denoising approaches may encounter difficulties in preserving fine image details while reducing noise. However, learning-based algorithms have demonstrated better capabilities in addressing this challenge compared to conventional techniques.

## 3. Topological Invariants of Image Data

This section describes the mathematical concepts required to create a topological loss function and explains how it can be exploited for image denoising.

Topological Data Analysis (TDA) has emerged as a powerful tool for analyzing complex datasets in various fields. It uses algebraic topology to study the shape and structure of data, including clusters, holes, and voids, which is particularly useful when dealing with data that may be high-dimensional or noisy. The TDA technique we employ here is called ‘*persistent homology*’, since it has been successfully applied to various problems, including genomics, material science, analysis of financial networks, and image processing [34,35,36,37,38,39].

In particular, the concept of persistent homology in TDA has shown promising results in topological image denoising [15]. It has been shown that, by applying a topological based loss function that operates in image patch space, noise removal can be enhanced while preserving essential topological features of the image.

Persistent homology can be used to deal with images by providing a robust and stable way to extract topological information from the image. By incorporating this information in the backpropagation process, persistent homology can assist in filtering out noise and extracting the underlying signal. The proposed loss function ensures that persistent homology features represented by the persistent diagram of the denoised output of the neural network are close to those of the clean image.

TDA was originally derived for continuous curves and manifolds, making it unsuitable for discrete objects such as images. To identify and compute topological invariants, which are properties of geometric objects that remain constant under certain transformations, the concept of a simplicial complex is introduced. A simplicial complex consists of simple shapes of various dimensions like points, lines, triangles, and tetrahedrons. These shapes are arranged by significance using a technique called filtration, which orders them from simpler to more complex. The basic idea is to begin with the most straightforward simplicial complex, which generally includes unconnected points or vertices, and then systematically include additional simplices to the complex in a controlled manner. This allows us to study the evolution of the topology of the object as it becomes more complex.

**Definition** **1** (Simplicial Complexes).
*Let V be a finite nonempty set, the elements of which are called vertices. A simplicial complex on V is a collection C of nonempty subsets of V that satisfies the following conditions:*
(*i*)
*∀v∈V, the set {v} lies in C; *
(*ii*)
*∀α∈C, and β⊆α, where β is also an element of C. α and β are called a simplex and a face, respectively.*



Conditions (*i*)–(ii) mean that any face or subset of vertices of a simplex in the collection is also in the collection and the intersection of any two simplices in the collection is either empty or a face of both simplices.

The *dimension of a simplex* α∈C is defined as dim(α)=|α|−1, the dimension of C is the highest dimension of constituent simplices.

In other words, a simplicial complex is a collection of simplices that fit together like puzzle pieces, where the boundaries of each simplex are shared with other simplices in the collection.

A *subcomplex* of a simplicial complex is a subset of its simplices that itself forms a simplicial complex. In other words, a subcomplex is a simplicial complex that is contained within another simplicial complex, or more formally, $C^{\prime }$ is called *a subcomplex of C*, if $C^{\prime }\subseteq C$ and $C^{\prime }$ is a simplicial complex itself. Intuitively, a subcomplex of a simplicial complex is a smaller “piece" of the original complex that retains its simplicial structure. For example, a triangle that is a face of a larger tetrahedron can be considered a subcomplex of that tetrahedron.

A simplicial complex is a general concept in topology that provides a structured way to study the topological properties of spaces. One specific type of a simplicial complex used in topological data analysis is the Vietoris–Rips complex.

**Definition** **2** (Vietoris–Rips Complex).
*Given a finite set of points P in a metric space (X,ρ) and a distance threshold ε>0, the Vietoris–Rips complex $V_{\varepsilon }\left(P\right)$ is a simplicial complex whose k-simplices correspond to subsets S⊆P with diameter diam(S) less than or equal to ε, i.e., $\mathrm{diam}\left(S\right)=\max_{x,y\in S}rho\left(x,y\right)\leq \varepsilon$. Formally, a k-simplex $S=\{p_{0},p_{1},\ldots ,p_{k}\}$ belongs to $V_{\varepsilon }\left(P\right)$ if and only if $\rho (p_{i},p_{j})\leq \varepsilon$ for all 0≤i,j≤k.*


In other words, the Vietoris–Rips complex is constructed from a set of data points by connecting those points that are within a certain distance threshold of each other. This type of complexes is useful for analyzing the shape and clustering of multi-dimensional data, revealing different levels of detail as ε changes.

There are other types of widely used complexes in topological analysis. One of them, a cubical complex, specifically caters to the data structure of digital images [40]. Cubical complexes provide a different approach by utilising the structure composed of axis-parallel cubes that encapsulate the geometry and relationships between cells of varying dimensions in a more grid-like manner.

However, although cubical complexes are useful for digital image data [40], they have an inherent higher-dimensional nature and a larger number of possible interactions between cells. This results in increased computational overhead and slower processing times compared to the relatively simpler pairwise distance calculations used in the Vietoris–Rips complex construction. Computation times for RGB image patches are shown in Figure 1. For example, for the 512 × 512 patch, the computation time for cubical complex (the computation time is the same for 0 and 1 dimensions due to the algorithm) is ∼5-times higher than for the 1-dim Vietoris–Rips and ∼170-times higher than for the 0-dim Vietoris–Rips complex (computed on Nvidia 2080 Ti). Due to this considerable limitation and the critical importance of the balance of computational speed and image patch size for deep learning-based denoising, throughout this paper, we will assume the use of Vietoris–Rips complexes.

Simplices in a simplicial complex are arranged by significance using a technique called *filtration*.

**Definition** **3** (Filtrations).
*A (n-)filtration of a simplicial complex is a sequence of subcomplexes, ordered by inclusion, such that each subcomplex is contained in the next. In other words, it is a sequence of simplicial complexes:*

(1)
$${\mathrm{FC}}_{0}\subset {\mathrm{FC}}_{1}\subset {\mathrm{FC}}_{2}\cdots \subset {\mathrm{FC}}_{n}=C$$



In TDA, while simplicial complexes are a convenient way to represent the connectivity and topological relationships within data, a filtration is used to provide a hierarchical representation of these relationship, revealing the evolution and persistence of topological features across different scales.

The purpose of filtrations is to capture the occurrence (“birth”) and disappearance (“death”) of topological-like homology classes [36], that can emerge and vanish as the parameter ε of the filtration $C_{\varepsilon }$ changes (for Vietoris–Rips complexes).

To visualise the changes in the topology of a simplicial complex over a range of filtration scales, a persistent diagram is created. The diagram displays each homology class [36] as a point, where the horizontal axis and the vertical axis represent the “birth” and “death” times, respectively.

**Definition** **4** (Persistent diagrams).
*A persistent diagram of a filtration F on an n-dimensional complex C is a collection of mappings $PD={\left\{PD_{k}\right\}}_{k=1}^{m<=n}$, each of which map every $i^{th}$ k-dimensional topological feature to a pair $\left(b_{i},d_{i}\right)\in R^{2}\cup \infty$:*

(2)
$$PD_{k}:\left(C,F\right)\to {\left\{b_{i},d_{i}\right\}}_{i\in I_{k}},$$

*where $b_{i}$ represents the appearance (“birth”) of the feature when $\varepsilon =b_{i}$ and $d_{i}$ represents its disappearance (“death”) when $\varepsilon =d_{i}$ (see Figure 2). When k=0, each point in the persistent diagram represents a connected component and when k=1, a point corresponds to a 1-dimensional holes in filtered subcomplexes of C.*


For image data, we calculate persistent diagrams on the filtrations of Vietoris–Rips complexes over the image intensity scale, tracking the occurrence and disappearance of topological features at different intensity values. For our task, we consider k=0 to represent the number of connected components and k=1 to represent the number of 1-dimensional holes in filtered subcomplexes of C. As this paper focuses on specific aspects of the topic and cannot accommodate a comprehensive discussion of concepts such as homology classes and groups, we refer the readers to [36] for more theoretical foundations and a deeper theoretical understanding of persistent diagrams.

## 4. Wavelet-Based Topological Loss Function

This work addresses the limitations of our previous approach [15] while still utilising topological information for denoising. Unlike our prior method, which extracted topological features from the patch space, we now compute them directly in the image spatial domain. Additionally, we introduce a wavelet-based mask to suppress artefacts in homogeneous areas.

### 4.1. Topological Loss Function

To construct a topological loss for a pair of ground truth and output images $\left(I^{C},I^{O}\right)$, we first calculate persistence diagrams $PD(I^{C})$ and $PD(I^{O})$ using intensity-based filtrations of Vietoris–Rips complexes calculated in the spatial dimension of the images. The topological distance between the two diagrams is then calculated using total persistence as a dissimilarity measure.

**Definition** **5** (Total Persistence). 
*p-total persistence of a persistence diagram PD is defined as a total sum of the lifespans of each point in a diagram:*

(3)
$$TPers\left(PD\right)=\sum_{x\in PD}{\left(d\left(x\right)-b\left(x\right)\right)}^{p},$$

*where d(x) and b(x) are “death” and “birth” values of x∈PD. For our experiments, we use p=1.*


Total persistence can be interpreted as the overall importance of topological features in a simplicial complex over a range of filtration scales.

Finally, we can use total persistence as the dissimilarity measure for the persistence diagrams space. We introduce the topological component of our future loss function, which we define as shown in Equation (Equation 4), where the absolute value is used instead of squaring to ensure robustness to outliers: (4)$$L_{top}\left(I^{O},I^{C}\right)=\left|TPers\left(PD\left(I^{O}\right)\right)-TPers\left(PD\left(I^{C}\right)\right)\right|.$$

A topological loss function based on persistent homology in the space of contrast patches sampled spatially from an image was shown to improve the image denoising task by Malyugina et al. in [15]. Unlike [15], in this work, we obtain topological information directly from the image spatial domain, leading to increased information on both local and global features. We combine it with a spatial loss to retain local spatial information. For the calculation of complexes, filtrations, and persistence diagrams, we used torch-topological (https://github.com/aidos-lab/pytorch-topological, accessed on 1 January 2025) package.

### 4.2. Texture Mask

In our experiments with $L_{top}$, defined in Equation (Equation 4), we observed that the topological loss component works better in textured areas or on edges, while sometimes creating undesired artefacts particularly in homogeneous regions.

Persistent homology inherently relies on intensity variations to capture topological features, which makes it especially sensitive to high-contrast regions. In such regions, strong intensity gradients define distinct topological features, such as connected components and loops, that emerge rapidly during the filtration process. These changes in the filtrations lead to a set of persistent features that dominate the persistence diagram due to the global nature of topological features. This can cause the loss function to predominantly focus on structural details in high-contrast areas while neglecting smoother regions. Such an imbalance may force the network to preserve noise-induced or non-essential details, resulting in artefacts during denoising.

To mitigate these issues, we incorporate a wavelet-based texture mask that selectively applies the topological loss in regions with strong textural content while reducing its influence in homogeneous areas. This ensures that the model benefits from topological constraints where they are most meaningful while avoiding unnecessary artefacts in textureless regions. Wavelets have been proven to be successful in capturing textual information in images through various applications, including texture classification [41], and image denoising with texture preservation [42]. The patterns and variations in the wavelet coefficients can provide information about the spatial frequency content and orientation of the texture.

The Discrete Wavelet Transform (DWT) decomposes a signal into a set of mutually orthogonal basis wavelet functions. To retrieve textured areas from the ground truth image $I^{C}$, we first apply 1-level decomposition with Haar base functions [43] and take only low-frequency bands, LL. This aims to ignore the areas with fine textural detail and noise [44], producing a smoother mask. Subsequently, we apply another DWT to LL and retrieve absolute values from the high-pass subbands HH,HV, and HD of LL, where HH, HV, and HD denote high-pass subbands in horizontal, vertical, and diagonal directions, respectively. Finally, we take mean values of these three level 2 DWT subbands, thus resulting in a *texture mask* (see Figure 3):$$Mask\left(I^{C}\right)=\mathrm{mean}\left\{d_{H}\right\}{|}_{H\in \{HH,HV,HD\}},$$
where$$d_{H}=|\;DWT_{LL}\circ DWT_{H}\circ {↑}_{4}\left(I^{C}\right)|,$$
and ${↑}_{4}$ denotes upscaling with the factor of 4 and ∘ is a function superposition.

Finally, to apply topological loss only to textured areas, we use the texture mask for image $I^{C}$, producing $Mask(I^{C})$ and combining it with pixelwise basic loss $L_{base}$, where $L_{base}=\ell_{1}$ or $\ell_{2}$:$$\begin{array}{c} L_{wvcomb}\left(I^{O},I^{C}\right)=L_{top}\left(I^{O},I^{C}\right)⊙Mask\left(I^{C}\right)+\\ \alpha L_{base}\left(I^{O},I^{C}\right)⊙\left(I-Mask\left(I^{C}\right)\right) \end{array}$$
where ⊙ is an element-wise product and α is a scalar hyperparameter. The respective method is shown in Algorithm 1.

Figure 4 shows the diagram of the proposed wavelet-based topological loss function. The restored image output of the denoising network is transformed into topological features, which are subsequently compared with those of the groundtruth, Ltop. The final loss function is a combination of Ltop and Lbase with the weighting determined by the textural information extracted from the wavelet domain.
**Algorithm 1:** Wavelet Topological Image Denoising**Input**: Noisy image IN, Groundtruth image IC, Trainable model fθ**Output**: Optimised parameter set θ^**Step 1: Wavelet Decomposition**;Perform wavelet decomposition on IC to obtain LL band wavelet coefficients; ILLC=DWTLL(IC)**Step 2: Texture Mask Calculation**;Calculate the texture mask; Mask(IC)=DWTH(ILLC)**Step 3: Acquire model output IO**;Apply model fθ to noisy image: IO=fθ(IN)**Step 4: Persistence Diagram Calculation**; Calculate the persistence diagrams PD(IC) and PD(IO) of IC and IO, respectively;**Step 5: Topological Loss Term Calculation**;Calculate the total persistence values TPers(IC) and TPers(IO) for PD(IC) and PD(IO), respectively;Calculate the topological loss component Ltop:Ltop(IO,IC)=|TPers(PD(IO))−TPers(PD(IC))|;**Step 6: Base Loss Term Calculation**;Calculate the base loss component Lbase as the ℓp loss between IC and IO;Lbase=||(IO,IC)||p,p=1or2**Step 7: Combined Loss Calculation**;Calculate the pixelwise mask-weighted topological loss to enforce the topological guidance in textured areas:Ltopmasked=Ltop⊙Mask(IC);and in remaining “plain” areas tune it down:Lbasemasked=Lbase⊙(1−Mask(IC));Calculate the combined wavelet loss Lwvcomb as the (α-weighted) sum of the mask-weighted losses with gain α;Lwvcomb(IO,IC)=Lbasemasked+αLtopmasked**Step 8: Find the parameters of the network, minimising combined loss**;θ^=argminθLwvcomb(fθ(IN),IC)**return** θ^;

## 5. Experiments and Discussion

### 5.1. Dataset

We trained and tested the models on the BVI-Lowlight dataset [15,45], which was collected to overcome the limitations of existing denoising datasets in terms of variations in noise levels and diverse content with varying textures. Unlike synthetic datasets, BVI-Lowlight captures complex spatial noise patterns, making it well-suited for testing our method’s ability to preserve texture. Its higher bit depth ensures a wider dynamic range and more accurate noise representation while preserving the details and textures in low light areas.

This dataset consists of 31,800 14-bit images captured from 20 scenes using two cameras, Nikon D7000 and Sony A7SII, with ISO settings ranging from 100 to 25,600 and 100 to 409,600, respectively. We ensured consistent lighting conditions by using non-flickering LED lights. Pseudo-ground truth images were generated by applying postprocessing techniques, including the removal of oversaturated pixels, intensity alignment, and image registration. Further details can be found in [15].

### 5.2. Training

In our experiments, we employed both traditional and state-of-the-art architectures for our baselines: (i) A residual-based denoising convolutional neural network (DnCNN) introduced in [10] by Zhang et al., which is a widely used method for benchmarking deep image denoisers. (ii) UNet, an encoder-decoder style fully convolutional network [46], which has been extensively used as a foundational framework for image denoising. (iii) RIDNet, a single-stage blind real image denoising network that incorporates a feature attention mechanism [11].

We compared the proposed loss with other denoising losses, including conventional losses $\ell_{1}$, $\ell_{2}$, and VGG loss (α is set to 0.99942857 as in [47]). In our experiments, we set topological loss term gain α to 0.004 as it demonstrated best performance on test sets. We also tested several combinations of losses and models.

All models presented in this paper were trained using 11K patches from the BVI-Lowlight dataset. The patches were sized 256 × 256, and we trained the models for 40 epochs. The initial learning rate was set to 0.0001, and the batch size was 16, which was limited by the memory capacity of our computing system.

### 5.3. Results

We assessed the performance of our method by training state-of-the-art denoising models with various loss functions and their combinations.

We evaluated image quality using Peak Signal-to-Noise Ratio (PSNR), Structural Similarity Index Measure (SSIM), and Learned Perceptual Image Patch Similarity (LPIPS). PSNR measures pixel-wise fidelity, defined as$$\mathrm{PSNR}=10\cdot \log_{10}\left(\frac{{\mathrm{MAX}}^{2}}{\mathrm{MSE}}\right),\;\mathrm{MSE}=\frac{1}{MN}\sum_{i=1}^{M}\sum_{j=1}^{N}{\left(I\left(i,j\right)-K\left(i,j\right)\right)}^{2}.$$
where I(i,j) and K(i,j) are pixel intensities of the original and denoised images, M×N is the image size, and MAX is the maximum possible image pixel intensity value.

SSIM assesses structural similarity by incorporating terms related to luminance, contrast, and correlation:$$\mathrm{SSIM}\left(I,K\right)=\frac{\left(2\mu_{I}\mu_{K}+C_{1}\right)\left(2\sigma_{IK}+C_{2}\right)}{\left(\mu_{I}^{2}+\mu_{K}^{2}+C_{1}\right)\left(\sigma_{I}^{2}+\sigma_{K}^{2}+C_{2}\right)}.$$
where $\mu_{I}$ and $\mu_{K}$ are mean intensities, $\sigma_{I}^{2}$ and $\sigma_{K}^{2}$ are variances, $\sigma_{IK}$ is the covariance, and $C_{1}$, $C_{2}$ are small constants to ensure stability.

LPIPS [48] is an image quality metric that computes feature-space distances from a pre-trained neural network (for our task, we used AlexNet) and is aligned more closely with human perception.

The results based on objective metrics are presented in Table 1 and the examples for subjective assessment are shown in Figure 5 and Figure 6. These results demonstrate improved performance in terms of objective metrics that correlate more strongly with subjective quality assessment [48].

The LPIPS metric shows the highest and most consistent improvement across all architectures when using the topological loss, demonstrating its effectiveness in enhancing perceptual quality. Even when combined with VGG loss, $L_{top}$ consistently outperforms conventional loss functions. We also have observed that perceptually, adding VGG loss in some cases helps to attenuate artefacts that can appear in untextured areas when using masked topological loss only.

For the RIDNET architecture, while PSNR for models trained with $L_{top}$ is slightly lower compared to those trained with $\ell_{1}$ or $\ell_{2}$, the LPIPS values remain significantly better, indicating superior perceptual quality. Notably, SSIM and PSNR are highest for RIDNET+$\ell_{2}$, suggesting that the model may require different weighting coefficients for the topological and VGG losses to fully optimize its performance. Examining the subjective results further in Figure 5 (rows 3 and 5), we can confirm that the RIDNET+$\ell_{2}$+$L_{top}$ (masked) model produces sharper textures and preserves fine details better than RIDNET+$\ell_{2}$. In particular, in regions with complex structures, such as circuitry and fabric, the masked topological loss reduces oversmoothing while maintaining perceptual consistency. This aligns with the improved LPIPS metric, suggesting that the model benefits from the topological loss in preserving high-level perceptual features without significantly altering SSIM or PSNR.

It is also worth noting that adding VGG loss term to a minimization criterion does not always improve the performance, comparing to models trained with basic loss only (see values for DnCNN with $\ell_{2}$). This further highlights the importance of careful loss balancing when integrating multiple perceptual and structural constraints.

Figure 5 shows that using a wavelet mask together with a topological term for the loss helps to reduce the occurrence of artefacts in homogeneous areas while preserving the texture and enhancing the contrast. For example, the denoised coin image using RIDNET with our proposed loss function exhibits sharper edges and better details compared to those denoised using $\ell_{2}$ alone or without a mask. Additionally, RIDNET with the unmasked loss function fails to completely remove color noise.

Figure 6 shows the denoised results using DnCNN. The results obtained by training it with a topological loss are shown with wavelet-based masks. Comparing the results in the 5th and 6th rows to those in the 3rd and 4th rows, our proposed masked topological loss produces sharper results with better fine details. Specifically, the fonts in the third column become more readable, such as ‘R61’. Without a topological loss, the VGG loss appears to remove the fine texture present in the top-right area of the images in the fourth column, which is the white area of the lama’s doll fur. More subjective results are available in Appendix A.

Overall, the evaluation of the results based on the BVI-Lowlight dataset, featuring real-world noise levels, has demonstrated the effectiveness of the proposed loss function in enabling denoising models to better learn both noise characteristics and the underlying signal. This, in turn, leads to enhanced contrast and preserved textural information in the denoised images.

In the presence of noise, the topological properties of an image can become distorted or disrupted. By incorporating topological invariants into the loss function, the algorithm can effectively capture and preserve the original structural features of the image while removing the noise. This approach helps to maintain the integrity of textured areas, and enhance the overall visual quality of the denoised image, as well as provide a valuable framework for improving the performance of existing image denoising models and combating common problems such as oversmoothing.

## 6. Conclusions and Future Work

In this paper, we introduce a novel method for image denoising that incorporates a wavelet-based topological loss function and utilizes textural information from the image wavelet domain. Our proposed loss function effectively captures noise characteristics while preserving image texture, resulting in denoised images with enhanced visual quality and preserved perceptual fidelity of the original content. The experimental results on real noise confirm the efficacy of our method, showing superior perceptual results and improved performance in objective metrics that are highly correlated with subjective quality assessment.

This study demonstrates that the proposed wavelet-based topological loss function improves low-light image denoising. However, there are still areas to explore. One direction is testing the loss function performance on different datasets with various noise patterns and scene types. Our experiments focus on the BVI-Lowlight dataset, which represents real-world low-light noise. Future research could extend this to datasets with different camera models and environments.

Another key aspect to investigate is the sensitivity of hyperparameters, especially the impact of the topological loss weight, α. We found an optimal value through experiments, but a more detailed study across a wider range of values could offer better insights.

Finally, since deep learning-based denoising models are evolving quickly, future work could also integrate our loss function into newer architectures, such as transformer-based denoisers or diffusion models. While we tested it with established models like DnCNN, UNet, and RIDNet, applying it to these newer approaches would further confirm its usefulness in modern image restoration.

By addressing these aspects in future research, the proposed loss function can be more comprehensively assessed in terms of its generalization capabilities, parameter robustness, and applicability to the latest state-of-the-art architectures for image denoising.

## Figures and Tables

**Figure 1 sensors-25-02047-f001:**
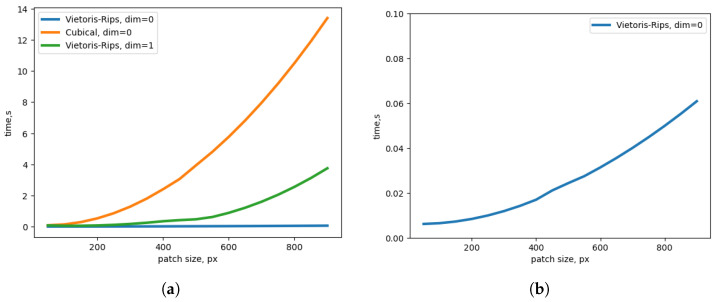
(**a**) Computation times of Vietoris–Rips complexes and cubical complexes of dimensions 0 and 1. (**b**) Computation times of the Vietoris–Rips complex of dimension 0.

**Figure 2 sensors-25-02047-f002:**
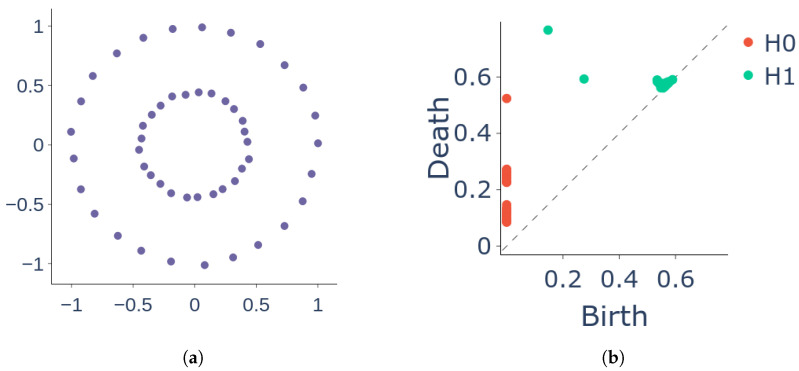
A persistent diagram PD(X) of a data cloud X shows the birth (represented by the *x* coordinate) and death (represented by the *y* coordinate) of topological features of different dimensions (0-dimensional and 1-dimensional features, denoted as H0 and H1, respectively). Each point in the diagram corresponds to one of these features. (**a**) Datacloud X. (**b**) Persistent diagram PD(X) for the datacloud X.

**Figure 3 sensors-25-02047-f003:**
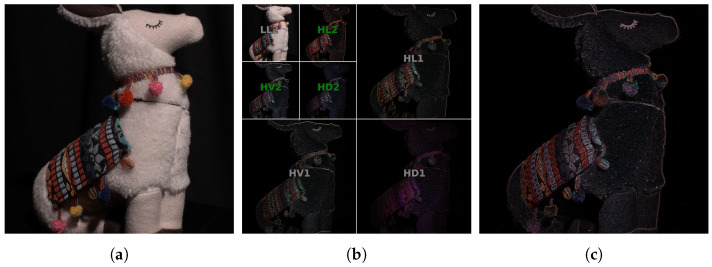
(**a**) Ground truth image $I^{C}$; (**b**) wavelet bands of 2-level decomposition $DWT(I^{C})$; (**c**) resulting wavelet mask $Mask(I^{C})$ built by averaging 2nd-order high-pass bands (green in (**b**)).

**Figure 4 sensors-25-02047-f004:**
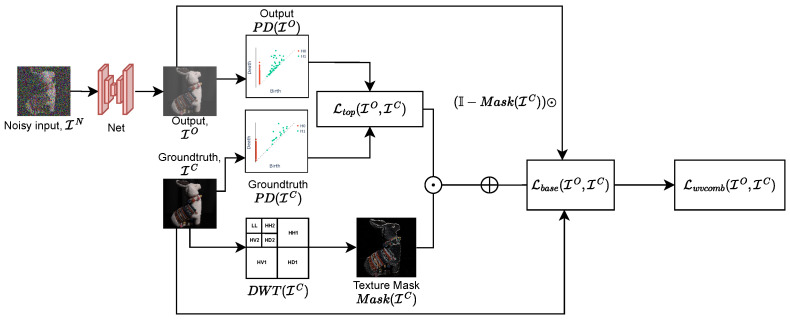
Wavelet topological loss calculation for a pair of images: output image $I^{O}$ and groundtruth $I^{C}$. First, the texture mask of $I^{C}$ is calculated based on wavelet decomposition of ground truth image $I^{C}$. The topological loss component $L_{top}$ is calculated as the absolute difference in total persistence values of diagrams $PD(I_{C})$ and $PD(I_{O})$. We also calculate $L_{base}$ ($l_{1}$ loss) to retain image spatial information. The resulting wavelet combined loss $L_{wvcomb}$ is calculated as pixelwise mask-weighted base loss $M\left(I^{C}\right)\times L_{top}$ and $L_{base}$.

**Figure 5 sensors-25-02047-f005:**
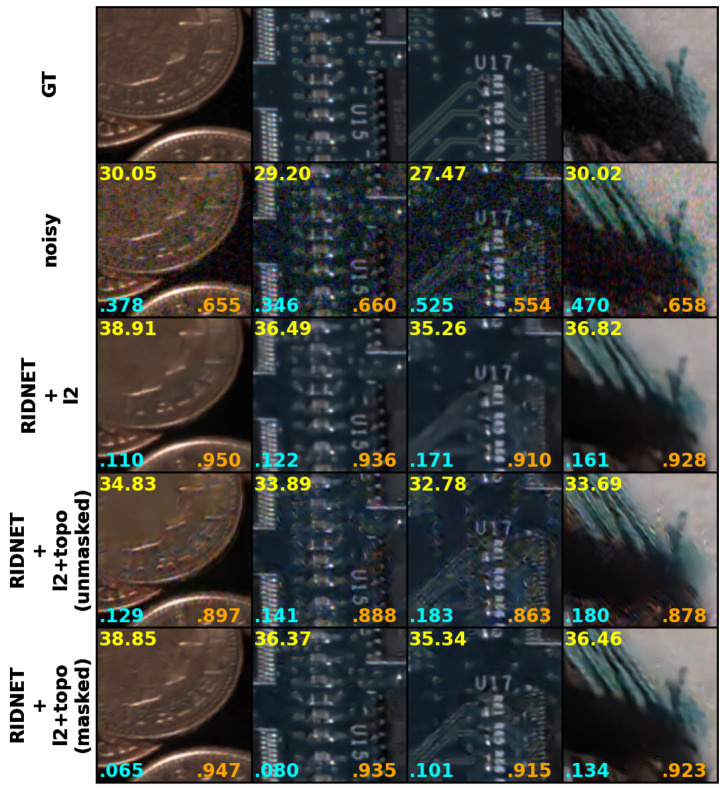
Using wavelet mask with $L_{top}$. **Top row**: Sample patches from dataset ground truth. **Second row**: Sample patches from noisy images with ISO varying from 160,000 to 400,000. **Third row**: Sample patches of the outputs from RIDNET trained with $\ell_{1}$ only. **Fourth row**: Sample patches of the outputs from DnCNN trained with topology loss (unmasked) combined with $\ell_{1}$. **Bottom row**: Sample patches of the outputs from DnCNN trained with topology loss (masked) combined with $\ell_{1}$. PSNR (yellow), LPIPS (cyan) and SSIM (orange) values are calculated per corresponding pairs. Note the preserved edges, enhanced contrast and improvement in LPIPS metric values when using masked topological loss compared to $\ell_{1}$ loss only. Using a wavelet texture mask also prevents the network from producing undesirable artefacts.

**Figure 6 sensors-25-02047-f006:**
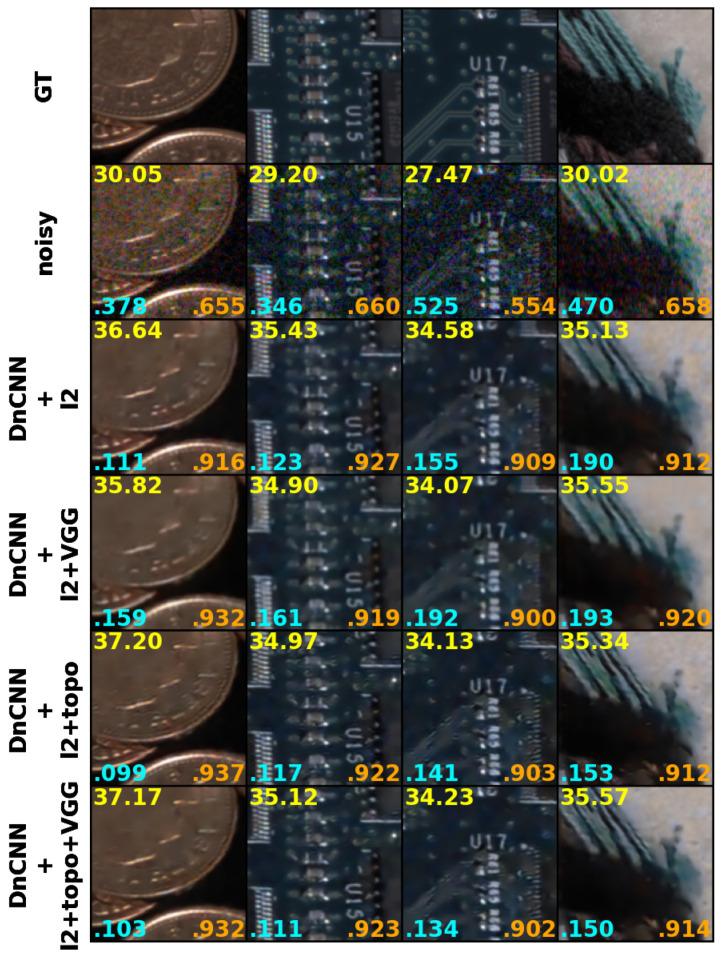
Results. **Top row**: Sample patches from dataset ground truth. **Second row**: Sample patches from noisy images with ISO varying from 160,000 to 400,000. **Third row**: Sample patches of the outputs from DnCNN trained with $\ell_{1}$ only. **Fourth row**: Sample patches of the outputs from DnCNN trained with VGG loss combined with $\ell_{1}$. **Fifth row**: Sample patches of the outputs from DnCNN trained with topology loss (masked) combined with $\ell_{1}$. **Bottom row**: Sample patches of the outputs from DnCNN trained with topology loss (masked) combined with VGG and $\ell_{1}$. PSNR (yellow), LPIPS (cyan) and SSIM (orange) values are calculated per corresponding pairs.

**Table 1 sensors-25-02047-t001:** Metrics values for the outputs of a corresponding model (DnCNN, RIDNET, or UNet) trained with (i) $\ell_{i}$ only (ii) combination of $\ell_{i}$ with VGG loss (iii) combination of $\ell_{i}$ with persistence-based loss $L_{top}$ (iv) combination of $\ell_{i}$ with persistence-based loss $L_{top}$ and VGG loss $L_{vgg}$. The first-place result for each model is highlighted in bold with a light green background, while the best overall result is highlighted in bold with a bright green background.

	LPIPS	PSNR	SSIM
Noisy	0.430	29.19	0.632
DnCNN+l1	0.172	35.26	0.918
DnCNN+l1+vgg	0.166	35.33	0.917
DnCNN+l1+topo	**0.129**	35.60	**0.923**
DnCNN+l1+topo+vgg	0.134	**35.66**	0.921
DnCNN+l2	0.154	35.54	**0.923**
DnCNN+l2+vgg	0.186	34.83	0.914
DnCNN+l2+topo	0.131	35.42	0.918
DnCNN+l2+topo+vgg	**0.124**	**35.55**	0.919
RIDNET+l1	0.119	36.97	0.934
RIDNET+l1+vgg	0.115	37.06	0.936
RIDNET+l1+topo	**0.096**	**37.24**	**0.938**
RIDNET+l1+topo+vgg	0.108	37.23	0.937
RIDNET+l2	0.098	**37.30**	0.937
RIDNET+l2+vgg	0.098	37.23	**0.939**
RIDNET+l2+topo	0.094	36.65	0.928
RIDNET+l2+topo+vgg	**0.089**	36.91	0.933
UNet+l1	0.180	35.36	0.916
UNet+l1+vgg	0.179	35.34	0.915
UNet+l1+topo	**0.136**	**35.68**	0.923
UNet+l1+topo+vgg	0.139	35.65	**0.924**
UNet+l2	0.171	35.64	0.919
UNet+l2+vgg	0.165	35.75	0.922
UNet+l2+topo	0.121	35.82	0.919
UNet+l2+topo+vgg	**0.116**	**35.88**	**0.923**

## Data Availability

The datasets generated and analysed during the current study are available from the corresponding author on reasonable request.

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
