# Peer review of "Wavelet-Based Topological Loss for Low-Light Image Denoising"

_sensors, 2025, doi:10.3390/s25072047_

Round 1
Reviewer 1 Report
Comments and Suggestions for Authors
The topic of this paper is highly relevant, as applying deep learning techniques for image denoising is a significant contribution. However, I have several concerns regarding your work:
- The paper requires improvements in English language clarity and readability.
- The deep network used in your study appears to be a novel aspect of your work. However, you have not provided any explanation of its structure. Please include a figure illustrating its architecture and clarify whether it is your own design or sourced from existing research.
- You compare conventional denoising methods with deep learning-based approaches, but you have not specified the metrics used to evaluate their performance. Please include the relevant equations.
- The final table should be placed in the Results or Discussion section rather than in the Conclusion.
- Consider adding a graphical representation to illustrate your method and approach.
- Use figures instead of tables to present your results more effectively.
- Compare your results with state-of-the-art methods to highlight the significance of your approach.
These improvements will enhance the clarity and impact of your paper.
Comments on the Quality of English Languagethe paper needs more explanation to be much clear, but the idea is very interesting
Reviewer 2 Report
Comments and Suggestions for Authors
During the review of the manuscript, I noticed that some parts bear a high degree of similarity to the previously published [Wavelet-based Topological Loss for Low-Light Image Denoising]. I hope you can recheck carefully to ensure the originality and unique value of the submitted content. Looking forward to your feedback.
Reviewer 3 Report
Comments and Suggestions for Authors
The improvement of the loss function previously described by the authors is the cornerstone idea of the manuscript. Since the main concept of the denoising loss function was covered in prior works, the novelty of the described methods is modest. However, the authors emphasize the further development of the loss function and a thorough evaluation of the denoising algorithm’s performance based on comparison of results obtained using popular neural net architectures. The main contribution lies in the aim of preserving texture information through the use of wavelet masks. This approach shows promise and has the potential to be applied in other image processing tasks, including data compression and cryptography.
Despite the high overall merit, the revision of the manuscript is required to address the following concerns:
-
The rationale behind creating the self-developed dataset is stated in prior work [15] and is associated with the goal of broadening image diversity and evaluating the impact of ISO settings. The presented manuscript does not demonstrate any dependencies between ISO and selected metrics. Since the analyzed scenes are separated into image fragments during performance evaluation in the presented manuscript, the greatest variety in images can be achieved by combining existing state-of-the-art datasets. Please add a rationale behind the usage of the self-developed dataset in the current work.
-
The authors are strongly advised to revise the arrangement of figures in the manuscript. Figure 1 is displayed between lines 142 and 143 and first mentioned in line 242, while Figure 2 appears between lines 216 and 217 and is mentioned in line 215. Figure 5 is mentioned before Figure 3, etc. The current arrangement of figures makes navigating through the manuscript text difficult.
Round 2
Reviewer 2 Report
Comments and Suggestions for Authors
The core of this manuscript is to verify the effectiveness of the new loss function. By combining topological features and wavelet transform, the experiments have demonstrated an improvement in the LPIPS metric. However, there are several potential issues as follows: 1. The experiments are solely based on a single dataset (BVI-Lowlight). In addition to the experiments on the BVI-Lowlight dataset, it is recommended to test the new loss function on other datasets with different characteristics, such as datasets with different scenes and noise distributions, to fully verify the generalization ability of the method. 2. For the key hyperparameters in the experiments, such as the topological loss term gain α, it is insufficient to only provide the optimal value. A hyperparameter sensitivity analysis should be conducted to show the impact of different α values on the model performance, enabling readers to better understand the role of this parameter and the basis for its selection. 3. Although the applications of topological invariants and wavelet transform are introduced in the paper, the author lacks an exploration of the in - depth theoretical basis. If possible, an explanation should be provided on why the topological loss function constructed based on the texture information in the wavelet domain can better adapt to different image contents, and a more in - depth interpretation should be given from the perspective of signal processing. Furthermore, in subsequent research, if the aim is to comprehensively demonstrate the universality and advantages of the new loss function on various models, then the failure to apply it to newly proposed models in the past two years can be regarded as a research limitation (the author only compared DnCNN (2017), UNet (2015), and RIDNet (2019)). This may affect the wide applicability of the research conclusions, especially in the field where new models are emerging continuously and showing outstanding performance. If the author wishes to more fully verify the effectiveness of the loss function, I suggest supplementing experiments on new models (network models proposed in 2023 and later). Overall, the article presents a relatively complete research content.
